# Perovskite multifunctional logic gates via bipolar photoresponse of single photodetector

Woochul Kim [1,2,7], Hyeonghun Kim [2,3,7], Tae Jin Yoo[4], Jun Young Lee[1,5], Ji Young Jo [1], Byoung Hun Lee[4], Assa Aravindh Sasikala [6], Gun Young Jung [1✉] & Yusin Pak [2✉]

The explosive demand for a wide range of data processing has sparked interest towards a new logic gate platform as the existing electronic logic gates face limitations in accurate and fast computing. Accordingly, optoelectronic logic gates (OELGs) using photodiodes are of significant interest due to their broad bandwidth and fast data transmission, but complex configuration, power consumption, and low reliability issues are still inherent in these systems. Herein, we present a novel all-in-one OELG based on the bipolar spectral photoresponse characteristics of a self-powered perovskite photodetector (SPPD) having a back-to-back $p^+$-i-n-p-$p^+$ diode structure. Five representative logic gates ("AND", "OR", "NAND", "NOR", and "NOT") are demonstrated with only a single SPPD via the photocurrent polarity control. For practical applications, we propose a universal OELG platform of integrated 8 × 8 SPPD pixels, demonstrating the 100% accuracy in five logic gate operations irrelevant to current variation between pixels.

[1] School of Materials Science and Engineering (SMSE), Gwangju Institute of Science and Technology (GIST), Gwangju 61005, Republic of Korea. [2] Sensor System Research Center, Korea Institute of Science and Technology (KIST), Seoul 02792, Republic of Korea. [3] School of Engineering Technology, Purdue University, West Lafayette, IN 47907, USA. [4] Department of Electrical Engineering, Pohang University of Science and Technology, Gyeongbuk 37673, Republic of Korea. [5] Electronic Materials Research Center, Korea Institute of Science and Technology (KIST), Seoul 02792, Republic of Korea. [6] Nano and Molecular Systems Research Unit (NANOMO), University of Oulu, Oulu 90750, Finland. [7] These authors contributed equally: Woochul Kim, Hyeonghun Kim. ✉email: gyjung@gist.ac.kr; yusinpak@kist.re.kr

Optoelectronic logic gates (OELGs) are receiving significant attention as crucial building block of future integrated circuits for accurate and fast data processing[1–4]. Existing circuits or processors based on electronic logic gates would face limitations in computing extensive data sets which are expected to markedly increase in the fourth industrial revolution age, due to performance shortfalls in switching, operation, computing, and decision making/regenerating[5,6]. Thus, developing an innovative logic gate platform that can implement faster computation with less power consumption is imperative to fulfill upcoming new computing trends.

Accordingly, all-optical logic gates using light inputs and outputs have been actively studied to replace electronic binary adders, binary counters, decision circuits, optical processors, data encoders, and bit-pattern recognition circuits[7]. These optical logic gates are constructed based on intricate designs of optical components such as terahertz asymmetric demultiplexers, nonlinear interferometers, and semiconductor amplifiers[8,9]. Although the optical logic gate system has attractive merits, including good optical gain, saturation output power, and gain bandwidth, practical applications have yet to be reported due to its expensive and large components[10]. Moreover, the straight propagation of light is a considerable obstacle for building a highly integrated system since placing optical components in an integrated manner, i.e., the arrangement with many optical curvatures may cause a sizeable optical loss[11].

Therefore, an OELG, which has recently been in the spotlight due to its broad bandwidth, fast data transmission, and low cost[12,13], is considered a good candidate for replacing the existing logic gates (e.g., electronic and all-optical). A typical OELG consists of more than two semiconductor junctions that interact with the light inputs and release a Boolean electrical output[14,15]. The most well-studied OELGs are p-n heterojunction photodiodes or photodetectors (PDs) such as ZnO/CdSe[16], CuInS$_2$/TiO$_2$[17], and n-Si/MoO$_{3-x}$[18]. Since these PDs operate mainly based on unidirectional carrier transport, most studies have only reported a single logic gate (e.g., AND or OR) to the best of our knowledge. Thus, to implement multiple logic functions, there is a strong need to develop a layered semiconducting PD platform that can freely control the direction of photocarrier transport. For instance, Yang et al. reported the wavelength-induced dual-polarity phenomenon from a p-n heterojunction diode composed of n-type ZnO nanowires and p-type thermoelectric thin film (SnS or Sb$_2$Se$_3$)[19–22]. In addition, particular circuit configurations are required for operating each logic function, and the reliability of output discrimination is strongly influenced by electrical noise or output variation of each logic gate[14,23,24]. Therefore, the difficulty in integration, high power consumption, and inaccurate signal characteristics should be enhanced to employ them in practical OELG applications.

With the recent success of perovskite photovoltaics, many studies have focused on perovskite optical sensors and PDs. The crucial challenge of these devices is the achievement of high gain, low power, and miniaturization. In this regard, organometal halide perovskite PDs allowing high absorption coefficient[25], excellent quantum efficiency[26,27], and broad spectral responsivity[28–30] can be promising as the OELG material and device. The superior optoelectronic properties of organometal halide perovskite have been demonstrated in many studies[31–35]. Although high-speed broadband detection and superior responsivity characteristics have been developed, such perovskite PD platforms operating on the principle of unidirectional carrier transport cannot be used as an integrated OELG component that performs multiple logic functions.

This paper reports a novel all-in-one OELG system based on the bipolar spectral photoresponse characteristics of a self-powered perovskite photodetector (SPPD). Five representative logics ("AND", "OR", "NAND", "NOR", and "NOT") were demonstrated with only a single SPPD, which converts the optical inputs to electrical outputs. The pivotal keys for executing the multiple Boolean logics are: (1) a "back-to-back configuration" (i.e., p$^+$-i-n-p-p$^+$), based on two vertically stacked perovskite diodes; and (2) an "optical gate modulation" using visible and near-infrared light. The back-to-back SPPD demonstrated a sensitive photoresponse and a substantial on-off ratio. An eight-by-eight (8 × 8) logic array consisting of 64 SPPD pixels successfully executed the five basic logic gates at 100% accuracy, insusceptible to electrical noise or current variation between the pixels owing to bipolar photoresponse.

## Results

**Back-to-back p$^+$-i-n-p-p$^+$ structure of the SPPD.** The SPPD was composed of vertically stacked low-band perovskite (FA$_{0.5}$MA$_{0.5}$Pb$_{0.4}$Sn$_{0.6}$I$_3$, near infrared-Perov (NIR-Perov)) and high-band perovskite (MAPbI$_3$, visible-Perov (Vis-Perov)), as depicted in Fig. 1a. Phenyl-C$_{61}$-butyric acid methyl ester (PCBM) layer was inserted as an n-type semiconductor between Vis-Perov and NIR-Perov, as shown in the cross-sectional transmission electron microscope (TEM) image. The perovskite layers were prepared by a two-step method: (i) vapor-phase thermal evaporation of a metal halide film (PbI$_2$ for MAPbI$_3$ and PbI$_2$/SnI$_2$ for FA$_{0.5}$MA$_{0.5}$Pb$_{0.4}$Sn$_{0.6}$I$_3$); and (ii) conversion step to form the perovskite phase (see more details in "Methods"). The metal halide films formed by the two-step method were so resistive to solvent infiltration that the underlying layers were not chemically damaged, as reported in previous studies[36,37]. The ratio between organic and metallic cations (FA$^+$ (formamidinium):MA$^+$ (methylammonium) ≈ 1:1 and Pb$^{2+}$:Sn$^{2+}$ ≈ 4:6) for the low-band NIR-Perov was optimally determined to have high film crystallinity, efficient device performance, and narrow bandgap (Supplementary Fig. 1)[38,39]. The p-type PEDOT:PSS and Spiro-OMeTAD were used as bottom and top hole-transport layers connected to ITO and Au electrodes, respectively. The thickness of each perovskite layer (300 nm NIR-Perov and 200 nm Vis-Perov) was decided via a theoretical simulation, aiming a photoconductive gain ratio ($G_{940}/G_{625}$) of 1 for efficient current offset, which is beneficial for the tuning of the logic gate operation (Supplementary Fig. 2). All the layers were vertically stacked without notable wrinkles and defects at each interface.

The element distribution was investigated by scanning transmission electron microscopy (STEM) equipped with energy-dispersive X-ray spectroscopy (EDX), as shown in Fig. 1b. Pb (red) and I (pink) elements were distributed uniformly in both perovskite layers (NIR-Perov and Vis-Perov). Sn (green) was only observed within the NIR-Perov layer, confirming that no perovskite merging occurred between the layers. Wide X-ray photoelectron spectroscopy (XPS) demonstrated the atomic ratio of Pb and Sn in the NIR-Perov layer to be approximately 4:6, which is known as the ideal composition for NIR absorption (Supplementary Fig. 3). In the perovskite phase, Sn$^{2+}$ and Pb$^{2+}$ were the primary oxidation states, according to the narrow Sn 3$d$ and Pb 4$f$ XPS spectra (Supplementary Fig. 2b and c), implying the superior film quality of NIR-Perov layer without any oxidation of Sn$^{2+}$. Notably, the absence of a Sn$^{4+}$ peak substantiated that a thiourea additive (see "Methods") was an effective anti-oxidative agent for protecting Sn$^{2+}$ cations from the undesirable oxidation[40]. The results of UV-Vis-NIR spectroscopy (Fig. 1c) showed that the Vis-Perov layer (MAPbI$_3$) absorbed the entire visible light (400–750 nm) while the NIR-Perov layer (FA$_{0.5}$MA$_{0.5}$Pb$_{0.4}$Sn$_{0.6}$I$_3$) could absorb even the infrared light (400–1000 nm). The bandgaps of both perovskites were 1.67 and

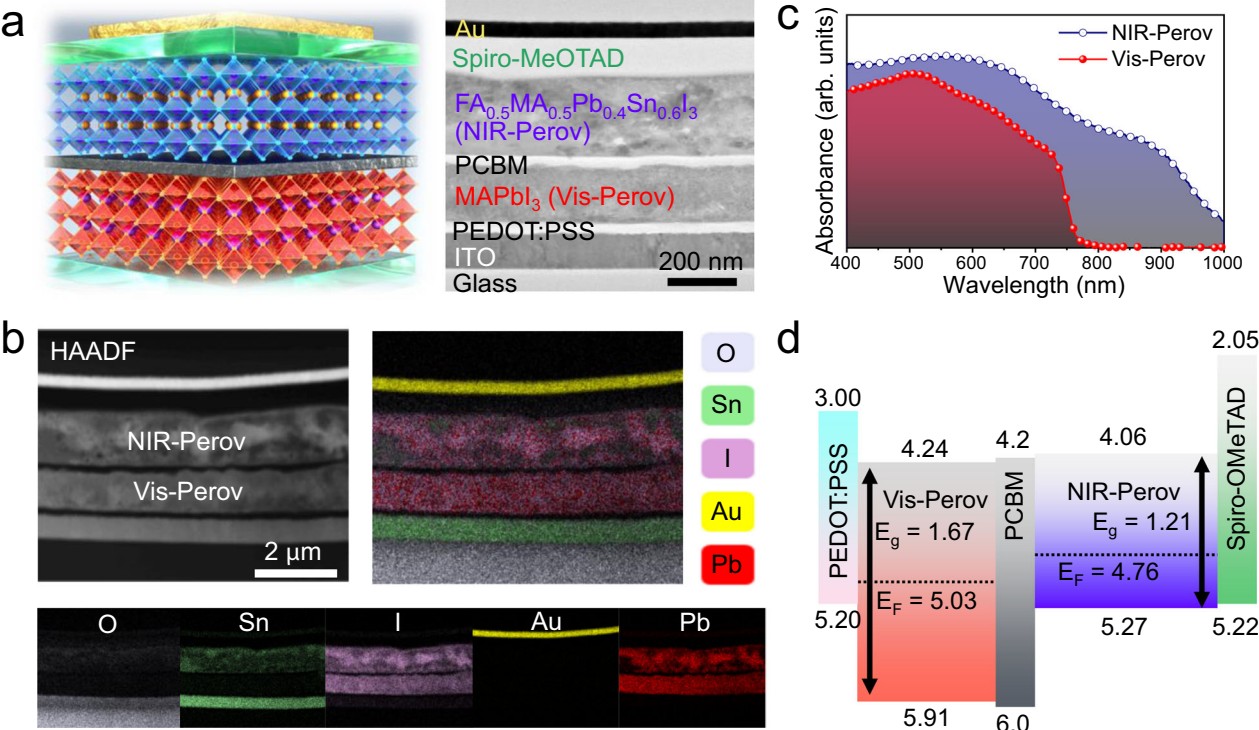

**Fig. 1 Characterization of the back-to-back SPPD. a** Schematic of vertically stacked SPPD and its corresponding cross-sectional TEM image. **b** STEM-HAADF image of the device and its corresponding elemental mapping images for O, Sn, I, Au, and Pb. **c** Absorption spectra of Vis-Perov and NIR-Perov layers. **d** Energy band diagram of the $p^+$-i-n-p-$p^+$ structure.

1.21 eV for the Vis-Perov and NIR-Perov layers, respectively, as calculated by the Tauc-plot method (Supplementary Fig. 4).

Figure 1d depicts the energy band diagram of SPPD, which was determined based on the result of UV photoelectron spectroscopy (UPS) (Supplementary Fig. 5) and relevant studies (for heavily doped p-type PEDOT:PSS[41], Spiro-OMeTAD[42], and n-type PCBM[41]). The Vis-Perov layer was intrinsic (i-type), and the NIR-Perov layer was lightly doped p-type, which were additionally supported by Hall measurements (Supplementary Table 1).

The PCBM could transfer electrons to either Vis-Perov or NIR-Perov layer at each heterojunction due to a low interfacial energy barrier. Density functional theory (DFT) calculations were conducted to obtain insight into the charge transport through the PCBM bridge (Supplementary Note 1). We found that additional density of states (DOS) below the conduction band were existing at the interface between perovskites and PCBM layers, which further facilitates the electron transfer to both perovskite layers (Supplementary Figs. 6 and 7). The symmetric band structure (back-to-back $p^+$-i-n-p-$p^+$ configuration) originating from the combination of two perovskite diodes provided a critical working principle that could control charge transfer in both directions (i.e., a bipolar spectral photoresponse) in response to irradiation conditions (e.g., wavelength and power density).

**Bipolar optoelectronic properties of the SPPD.** The back-to-back SPPD ($p^+$-i-n-p-$p^+$) showed bipolar spectral photoresponse depending on irradiation condition, which was quite different from a single diode (e.g., p-i-n or pn structure), yielding only unipolar photoresponse. The SPPD generated currents with opposite polarities depending on the visible (530 nm) and NIR (940 nm) irradiation, as shown in Fig. 2a. Open circuit voltages ($V_{OC}$) were –0.22 V for 530 nm and +0.09 V for 940 nm at a power density of 1 mW cm$^{-2}$. Figure 2b exhibits three on/off

cycles of SPPD at zero voltage. The responsivity values for the positive and negative photocurrents were 5.8 mA W$^{-1}$ for 530 nm and 8.7 mA W$^{-1}$ for 940 nm irradiation (Fig. 2b). The rise/decay times of SPPD with an active area of 0.04 mm$^2$ were 62/680 μs for 530 nm and 40/72 μs for 940 nm (Supplementary Fig. 8). Even though there is still an apparent gap compared to the state-of-art level (e.g., hundreds of ns, Supplementary Table 2), it can be enhanced using a high-crystalline perovskite film with further downsizing of the active area.

The bipolar spectral photoresponse of SPPD was investigated by illumination from 400 to 1000 nm (Fig. 2c). The responsivity was maximum at 500 nm, which was gradually decreasing with increasing the wavelength, and became zero at ~750 nm. Beyond 750 nm, the responsivity dramatically increased with opposite polarity. The zero responsivity point (750 nm) can be varied with the perovskite and intermediate components. Based on the spectral responsivity results, photoconductive gain ($G_\lambda$) and detectivity ($D_\lambda$) values were calculated as shown in Supplementary Fig. 9. The $G_\lambda$ values were 0.02 and 0.008 for 530 and 940 nm light illumination, respectively. The corresponding $D_\lambda$ values were calculated as 1.89 and 1.27 × 10$^{11}$ Jones, respectively.

The SPPD retained its initial responsivity for 3000 pulses (5 s on/5 s off) under ambient condition as shown in Supplementary Fig. 10. For long-term stability testing, the SPPD was stored in dry-air condition for 1 year. The saturated on-current was reduced to 90% and 50% of the initial value under 530 and 940 nm, respectively (Supplementary Fig. 11). The relatively significant reduction under the NIR light can be attributed to the instability of Sn cations in the air[42,43].

To theoretically evidence the bipolar spectral photoresponse, we performed an optical field simulation. A charge generation rate ($C_R$), occurring along a coronal plane of the back-to-back structure, was simulated under monochromatic irradiation

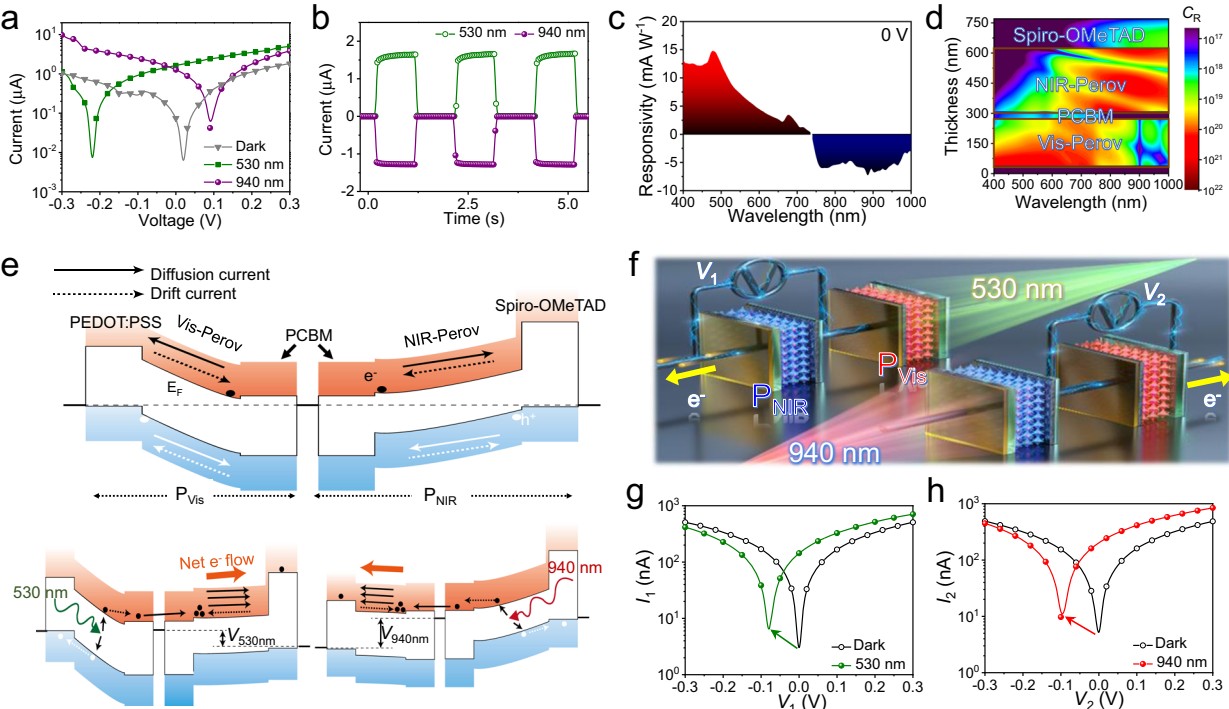

**Fig. 2 Optoelectronic properties of the SPPD. a**, **b** *I–V* curves and steady-state currents at zero bias under dark and irradiation conditions (530 and 940 nm lights at 1 mW cm$^{-2}$). **c** Bipolar spectral photoresponse of the SPPD as a function of wavelength from 400 to 1000 nm at a 10 nm interval. **d** Charge generation rate ($C_R$) distribution within the p$^+$-i-n-p-p$^+$ structure vs. wavelength of incident light. **e** Energy band diagrams of the SPPD (P$_{Vis}$ and P$_{NIR}$ serially connected through PCBM); under equilibrium state (top), 530 nm light irradiation onto the P$_{Vis}$ (bottom-left), and 940 nm light irradiation onto the P$_{NIR}$ (bottom-right). **f–h** Experiments for substantiating the working mechanism of (**e**). **f** Schematic illustration of equivalent circuits for the separate perovskite diodes (P$_{Vis}$ and P$_{NIR}$) connected with an electrical wire. *I–V* curves of **g** P$_{NIR}$ in response to the 530 nm light irradiation onto the P$_{Vis}$ and **h** P$_{Vis}$ in response to the 940 nm light irradiation onto the P$_{NIR}$.

(400–1000 nm). The $C_R$ was calculated as follows:

$$C_R = \frac{2\pi\varepsilon_0 nk}{h}\|\mathbf{E}\|^2, \tag{1}$$

where $\varepsilon_0$ is the permittivity of free space, $h$ is the Planck constant, $n$ and $k$ are the real and imaginary parts of the refractive index, and $\mathbf{E}$ is the optical field. The optical field was calculated based on the transfer matrix method[44] (Supplementary Fig. 12) using complex refractive indices and film thicknesses of components (Supplementary Fig. 13). Figure 2d exhibits the distribution profile of $C_R$ as a function of the wavelength. The $C_R$ intensities of two perovskites (NIR-Perov and Vis-Perov) at around 750 nm were approximately equivalent such that they readily offset each other, resulting in the net-zero photocurrent coinciding with Fig. 2c. The Vis-Perov layer demonstrated an intense (red) $C_R$ field in the visible range (400–750 nm), which weakened (blue) beyond 750 nm. In contrast, the $C_R$ field in the NIR-Perov layer strengthened from 750 nm. The main advantage of back-to-back configuration was that the offset wavelength could be readily tuned by diode materials and stacking order of perovskite materials, implying versatile applications in future optical and optoelectronic sensor chips.

The energy band diagram of SPPD could be represented as two separate diodes connected in series: p$^+$-i-n (P$_{Vis}$: PEDOT:PSS/Vis-Perov/PCBM); and n-p-p$^+$ (P$_{NIR}$: PCBM/NIR-Perov/Spiro-OMeTAD) as depicted in the upper scheme of Fig. 2e. The large difference in Fermi levels between the n-type PCBM and two heavily doped p-type layers (PEDOT:PSS and Spiro-OMeTAD) produced built-in potentials that attracted charges to the PCBM. More details on the built-in potential and energy band-bending are explicated in Supplementary Fig. 14. Without external stimuli

(e.g., light or electrical bias), the diffusion current driven by the carrier concentration gradient and the drift current by the potential gradient were equivalent, resulting in a net-zero current. However, as shown in the lower scheme of Fig. 2e, the balance between the diffusion and drift current was broken under illumination. Under 530 nm, the electrons generated at the Vis-Perov (P$_{Vis}$) layer diffused and piled up at the interface between PCBM and NIR-Perov (P$_{NIR}$) layer. When enough electrons were accumulated at the interface, an electrical potential ($V_{530nm}$) was built across the P$_{NIR}$ which cancel out the inherent built-in potential, thus generating a net electron diffusion toward the Spiro-OMeTAD layer without an external bias. Likewise, under 940 nm, the increased electron concentration gradient induced a potential ($V_{940nm}$) across Vis-Perov layer, leading to a net photocurrent. It is noteworthy that when the PCBM layer was not inserted, the photocurrent flowed unidirectionally from Vis-Perov to NIR-Perov as shown in Supplementary Fig. 15.

To experimentally prove the aforementioned optoelectronic mechanism that a current (or potential) is generated in a diode (not illuminated) by illuminating the other electrically connected diode, we fabricated a series circuit consisting of two separate perovskite diodes (P$_{Vis}$ and P$_{NIR}$), as depicted in Fig. 2f. Only one diode was irradiated with photo-active light, while the other was deliberately blocked from it. As shown in Fig. 2g, for the 530 nm irradiation onto the P$_{Vis}$, the P$_{NIR}$ produced a significant current of ~110 nA at 0 V owing to a generated electrical potential ($V_{OC}$ shift of −0.08 V). Similarly, under 940 nm irradiation onto the P$_{NIR}$, the $V_{OC}$ of P$_{Vis}$ shifted (−0.1 V), producing a reverse current of ~120 nA (Fig. 2h). These results demonstrated the self-powered SPPD and the necessity of integrated back-to-back SPPD (p$^+$-i-n-p-p$^+$) for enabling the bipolar spectral photoresponse.

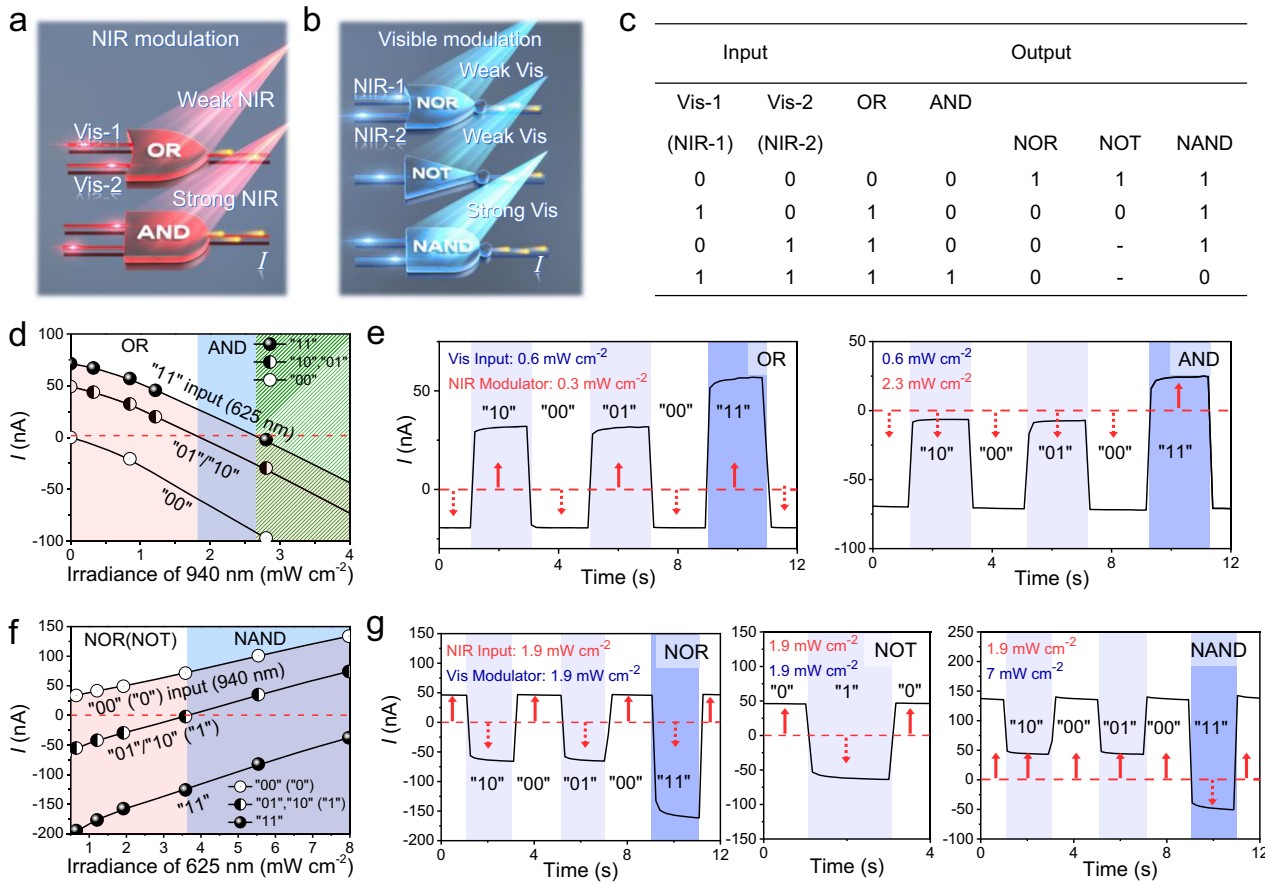

**Fig. 3 Demonstration of the OELGs via the single SPPD. a, b** Symbolic schematics of five OELGs executed with single SPPD. **a** "OR" and "AND" gates using two visible light inputs (625 nm) under NIR light (940 nm) gate modulations; **b** "NOR", "NOT", and "NAND" gates using NIR light inputs (940 nm) under visible light (625 nm) gate modulations. *I* is an output photocurrent. **c** Truth table of the five OELGs for the four input combinations. **d-g** Optoelectronic logic gate operations. **d** Photocurrent output via visible light (625 nm) inputs depending on NIR irradiance (940 nm). **e** Transient photocurrent curves of "OR" and "AND" gates in response to visible inputs (0.6 mW cm$^{-2}$) under NIR modulation of 0.3 and 2.3 mW cm$^{-2}$, respectively. **f** Photocurrent output via NIR (940 nm) light inputs depending on visible irradiance (625 nm). **g** Transient photocurrent curves of "NOR", "NOT", and "NAND" gates in response to NIR inputs (1.9 mW cm$^{-2}$) under visible modulation of 1.9, 1.9, and 7 mW cm$^{-2}$, respectively.

**Demonstration of five basic OELGs.** Based on the bipolar spectral photoresponse of SPPD, we propose a novel OELG system that can execute five basic logics of "OR", "AND", "NAND", "NOR", and "NOT" with only single SPPD (Fig. 3a, b). The key factor for tuning logic functions in this system is the "optical gate modulator", which involves irradiation using different wavelengths and intensities. Delicate tuning of logic functions via the light intensity modulation for the opposite photocurrent-offset can be most efficient when the two light wavelengths have similar photoconductive gains. As the 625 nm light revealed a well-matched photoconductive gain (0.007) with that (0.008) in case of the 940 nm light (Supplementary Fig. 9a), hereafter, we utilized these two light sources for the following logic gate operations. Prior to the use of 625 nm, we confirmed that the 625 nm irradiation generated the opposite photocurrents compared to the 940 nm case (Supplementary Fig. 16). Figure 3c shows the electrical logic outputs of the SPPD-OELG in response to the optical inputs ("00", "10", "01", and "11"), in which "0" and "1" signify "on" and "off" light input state, respectively. To the best of our knowledge, this is the first study that has successfully demonstrated five basic logic gates with a single device (SPPD) without any change in the circuit configurations. To emulate the "AND" and "OR" gates (Fig. 3d), we investigated the photocurrents of SPPD in response to visible light (i.e., 625 nm at 0.6 mW cm$^{-2}$) input combinations ("00", "01", "10", and "11") under varying

NIR irradiances (940 nm). Here, the NIR light (gate modulator) was irradiated simultaneously with the visible light input, producing a negative offset photocurrent against the positive current by the visible light input. The red dashed line in Fig. 3d represents the fiducial level (0 nA), which set the output state as "1" and "0" for the values above and below it, respectively.

Under low NIR irradiance (<1.8 mW cm$^{-2}$), if either input was true ("01" or "10"), the output photocurrent always exceeded the fiducial level positively. The negative photocurrent below 0 nA was obtained only for the "00" input, executing the "OR" gate. Meanwhile, under the NIR irradiance from 1.8 to 2.6 mW cm$^{-2}$, the "AND" gate was implemented to have "1" output state only when both inputs were true ("11"). The current levels of "10" and "01" inputs completely dropped below 0 nA by the NIR-modulated negative offset photocurrent. The transient curves (Fig. 3e) show all the possible input scenarios for both the "OR" and "AND" gates. The upward and downward arrows represent the "1" and "0" output states, respectively. For the "OR" gate under the NIR irradiance of 0.3 mW cm$^{-2}$, all the input combinations except for "00" input induced a positive photocurrent above 25 nA, corresponding to "1" output state. Interestingly, the "OR" gate promptly was transformed into the "AND" gate when the NIR irradiance increased to 2.3 mW cm$^{-2}$, which produced a positive photocurrent only for the "11" input. This tuning technique (OR↔AND) using the NIR modulation was reversible and repeatable.

Furthermore, the unique spectral photoresponse of SPPD also allowed to execute inverting logic gates, such as "NAND", "NOR", and "NOT". Figure 3f shows the photocurrents of SPPD under NIR inputs (i.e., 940 nm at 1.9 mW cm$^{-2}$) as a function of the visible irradiance (625 nm). In contrast to the previous NIR gate modulation, the photocurrent was increased with the irradiance of visible modulation. Under low visible irradiance (<3.6 mW cm$^{-2}$), the photocurrents were smaller than the fiducial level (0 nA) if one ("10" or "01") or both ("11") NIR inputs were applied. However, in case of "00" NIR input, a positive photocurrent was generated, corresponding to either the "NOR" or "NOT" logic gate. On the other hand, as the visible irradiance increased above 3.6 mW cm$^{-2}$, the photocurrent was positively converted except for the case of "11" NIR inputs, operating the "NAND" gate.

Figure 3g shows the OELG operation for implementing the three inverting logic gates. Visible modulation irradiances of 1.9 and 7 mW cm$^{-2}$ were used for "NOR" (and "NOT") and "NAND", respectively. Under a visible irradiance of 1.9 mW cm$^{-2}$, the output state of "1" (46 nA) was obtained only for the "00" NIR input (−64 nA for the "01" and "10" inputs and −160 nA for "11" input), demonstrating the "NOR" gate. In addition, at the same modulating condition, the "0" and "1" NIR single inputs revealed "1" (46 nA) and "0" (−64 nA) outputs, respectively, executing the "NOT" gate. Under 7 mW cm$^{-2}$ irradiance, a negative photocurrent was produced only by the "11" input, executing the "NAND" gate. In addition, we also confirmed logic gate operations using a pair of 530 and 940 nm light sources after precisely adjusting the irradiance of 530 nm light (Supplementary Fig. 17).

**SPPD array platform for advanced OELG applications**. For real-world applications, it is imperative to construct integrated SPPD-OELG platform and investigate its accuracy to ensure the practical chip-level feasibility. Figure 4a shows an eight-by-eight (8 × 8) crossbar-type SPPD array. Seven stacked layers (PEDOT:PSS/NIR-Perov/PCBM/Vis-Perov/Spiro-OMeTAD/MoO$_3$) were sandwiched between the top (transparent ITO) and bottom (Au) electrodes (see more details in "Methods"). The crossed area (dotted square) between the two electrodes served as a single-pixel logic gate with an active area of 200 × 200 μm$^2$; 64 pixels were integrated in total (Fig. 4b). Three-dimensional bar charts in Fig. 4c demonstrate all the outputs for the five logic gates obtained from the 64 pixels. The red and blue bars represent the positive and negative photocurrents, respectively, with reference to the fiducial level of 0 nA (gray face). All the corresponding pixels yielded the outputs with 100% accuracy in all five logic gates, irrelevant to the calculated coefficient of photocurrent variation (=standard deviation/average, 0.36–0.66) which affects the uniformity of output signals.

Even after downsizing the pixel to 50 × 50 μm$^2$, the SPPD array succeeded in yielding all the logic gate operations with 100% accuracy. The photocurrent was reduced with decreasing the active pixel size, but the output polarity was maintained in every pixel (Supplementary Figs. 18–20). The superior output reliability of SPPD array platform was attributed to our back-to-back SPPD structure that distinguished true or false output state based on the polarity of photocurrent rather than the difference between the unipolar current and fiducial level, normally used in other conventional logic gate devices.

Notably, from a practical perspective, this approach is an advancement over the reported optical and optoelectronic logic gates regarding the on/off ratio (theoretically infinite in this work, Supplementary Table 3), which is the key factor for precise output decision. Furthermore, the back-to-back SPPD is advantageous to multi-logic functionality with a single device, circuit compactness,

and high spatial efficiency, compared to all kind of logic gate devices.

## Discussion
We reported all-in-one OELG based on the SPPD to execute multi-logic gates that can build advanced logic gate circuits or processors. The pivotal point is the back-to-back SPPD structure (p$^+$-i-n-p-p$^+$) consisting of the low-band and high-band perovskites. The photocurrent polarity of SPPD could be deliberately tuned by optical gate modulation using visible (625 nm) and NIR (940 nm) lights. Thus, the output state of "1" or "0" was determined by the positive or negative output photocurrent, respectively. The single SPPD succeeded in configuring five logics: "AND", "OR", "NAND", "NOR", and "NOT". The output discrimination of SPPD-OELG was totally based on the polarity of photocurrent rather than the difference between unipolar photoresponse and fiducial level, improving the accuracy and reliability of logic gate device significantly irrelevant to current variation or electrical noise. The OELG array platform involving 64 SPPD pixels executed all the five logic gates at 100% yield without any errors.

The data processing under specific instructions (e.g., add, multiply, or count) could be implementable with a combinational circuit of multiple SPPD-OELGs in a single chip, which is much spatially and costly efficient compared to the conventional logic circuit based on electronic transistors, potentially advancing to future applications for optical computing, optical communication, and logic memory. In short-term view, this development can be applicable to light-fidelity (Li-Fi) transmission[21], security circuits[45], and healthcare sensors[46], utilizing the distinguished optoelectronic output states based on the photocurrent polarity.

## Methods
**Fabrication of the SPPD**. For the preparation of MAPbI$_3$, a PbI$_2$ film was thermally deposited on the PEDOT:PSS-coated ITO substrate, and MAI solution (50 mg mL$^{-1}$ in IPA) was spin-coated on the sample at a rate of 3500 rpm for 45 s. The PbI$_2$ and MAI layers were converted into the MAPbI$_3$ at 100 °C for 10 min. As an electron transport layer, a PCBM solution (20 mg mL$^{-1}$ in chlorobenzene) was spin-coated at 2000 rpm for 40 s and dried at 100 °C for 10 min. For the formation of FA$_{0.5}$MA$_{0.5}$Pb$_{0.4}$Sn$_{0.6}$I$_3$ perovskite layer, a Pb$_{0.4}$Sn$_{0.6}$I$_2$ composite film was deposited on the PCBM by co-evaporation of PbI$_2$ (0.8 Å s$^{-1}$) and SnI$_2$ (1.2 Å s$^{-1}$, with 20 mol% of SnF$_2$ additive). The film was converted into FA$_{0.5}$MA$_{0.5}$Pb$_{0.4}$Sn$_{0.6}$I$_3$ by spin-coating FA$_{0.5}$MA$_{0.5}$I solution (50 mg mL$^{-1}$ in IPA with 50 mol% of thiourea additive) under 3500 rpm for 45 s followed by an annealing process at 100 °C for 10 min. As a top hole-transport layer, 0.06 M Spiro-OMeTAD dissolved in chlorobenzene (28.8 μL of 4-tert-butylpyridine and 17.5 μL of 1.8 M Li-TFSI in acetonitrile) was spin-coated at 2000 rpm for 20 s. Finally, a gold electrode was deposited by thermal evaporation. The entire processes were conducted in a nitrogen-filled glove box.

**Fabrication of the SPPD-OELG array**. First, the patterned Au electrodes were thermally deposited onto a SiO$_2$/Si substrate with a shadow mask. The organic and perovskite building blocks were fabricated onto the Au electrode in the following sequence with the same methods above: PEDOT:PSS/FA$_{0.5}$MA$_{0.5}$Pb$_{0.4}$Sn$_{0.6}$I$_3$/PCBM/MAPbI$_3$/Spiro-OMeTAD. A transparent ITO electrode was deposited by RF sputtering at 50 W under Ar gas at a pressure of 5 mTorr. To prevent physical damage to the underneath organic layer during the sputtering process, we deposited a MoO$_3$ layer ($t \approx 30$ nm), serving as a buffer layer, on the Spiro-OMeTAD using a thermal evaporator.

**Material and device characterization**. Cross-sectional TEM images of the SPPD were observed using Tecnai G2 F30 S-Twin microscope operated at an acceleration voltage of 300 kV. Cross-sectional samples ($t \approx 100$ nm) for the TEM measurement were obtained using focused-ion beam (FIB) system (HITACH NX5000). Elemental analyses of the perovskite films were performed by energy-dispersive spectrometer (EDS) mode of TEM with 136 eV resolution and X-ray photoelectron spectroscopy (XPS, NEXSA) with Al Kα source. Work function and binding energy of perovskite films on Au substrate were measured using ultraviolet photoelectron spectroscopy (UPS, NEXSA) with He (I) and He (II) source after Ar ion milling for 30 s. The optical absorbance was measured using a UV-visible spectrometer (AvaSpec Spectrometer, Jinyoung Tech., Inc.). The optical images of perovskite photodetector array were observed by Olympus BX51 microscope. The photocurrent was measured using a Keithley 4200 source meter under illumination of

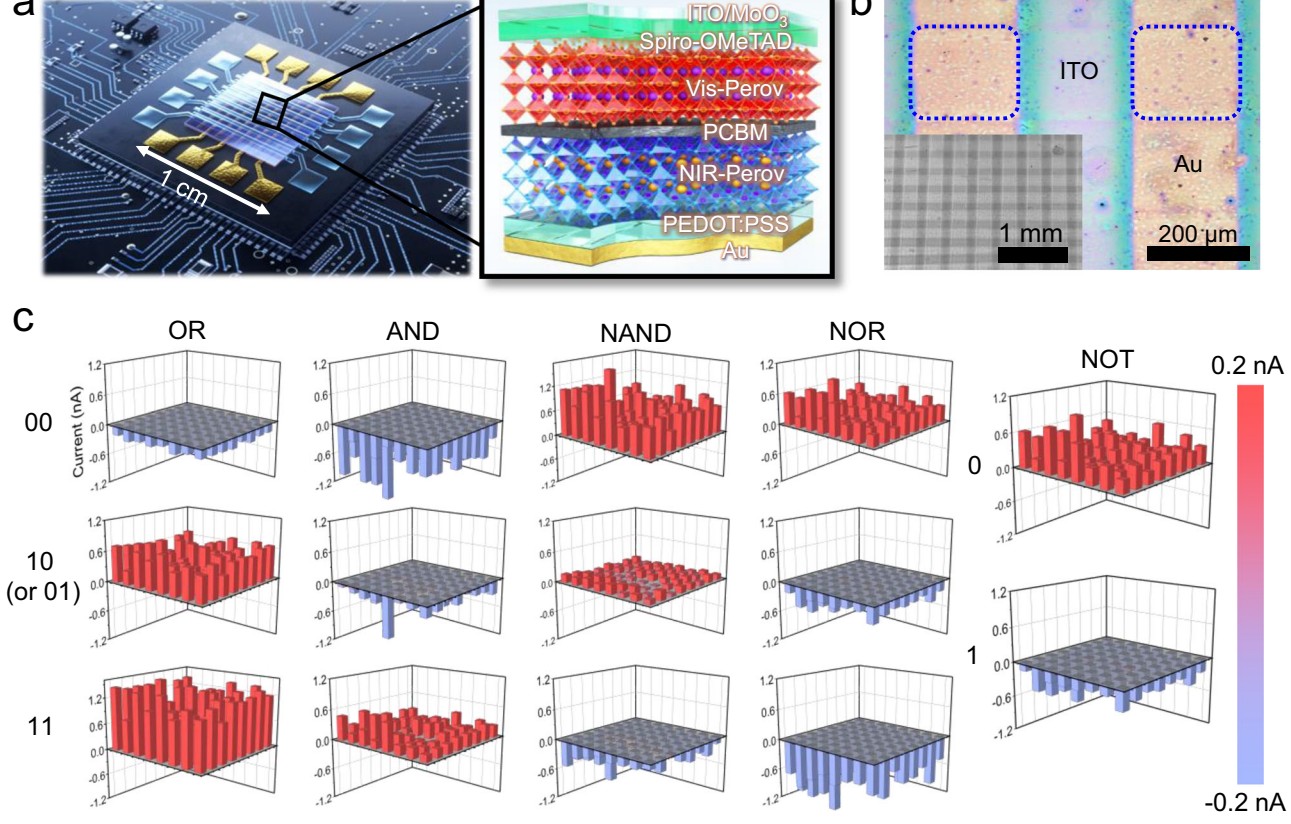

**Fig. 4 Logic gate operation of the SPPD array. a** Schematic of the (8×8) crossbar-type SPPD array platform (left) and cross-sectional illustration (right) of a single SPPD. **b** Optical microscopy image of the SPPD array. The inset of **b** shows a low magnified image. **c** Three-dimensional bar charts for all the outputs ("OR", "AND", "NAND", "NOR", and "NOT") obtained from the 64 pixels. The red and blue bars show clear bipolar spectral photoresponses of all the pixels with reference to the fiducial level of 0 nA (gray face).

LEDs (530, 625, and 940 nm, Mightex) and monochromatic light from xenon lamp (66902, Newport) equipped with monochromator (Mmac-200, Spectro). Rise and decay times were measured using an oscilloscope (Tektronix, DPD4014B) equipped with a low-noise current preamplifier (Stanford Research, SR570).

### Data availability
The data that support the plots within this paper and other findings of this study are available from the corresponding author upon reasonable request.

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

## Acknowledgements

W.K., G.Y.J., and Y.P. acknowledge the National Research Foundation of Korea (NRF) grant funded by the Korea government (MIST and MOE) (No. NRF-2019R1A2B5B01070640, 2020M3H5A108110412, 2019M3E7A1113097) and the KIST Institutional grant (2E31271). A.A.S. acknowledges the CSC-IT Center for Science (Finland) for the computational resources.

## Author contributions

W.K. and H.K. contributed equally to this work. W.K. and H.K. conceived the idea, carried out the experiments, and wrote the manuscript. T.J.Y., J.Y.L., B.H.L., and J.Y.J. contributed to the discussion on device fabrication and characterization. A.A.S. contributed to the analytical and computational analysis. G.Y.J. and Y.P. supervised and coordinated the entire works. All authors discussed the final manuscript, commented, and revised the submitted manuscript.

## Competing interests

The authors declare no competing interests.
