## [Peer Review File · Nature Communications]

Perovskite multifunctional logic gates via bipolar photoresponse of single photodetectorREVIEWER COMMENTS

Reviewer #1 (Remarks to the Author):

Major revisions.

In this work, the authors reported multifunctional logic gates which was constructed based on bipolar photoresponse from single photodetector. This is an interesting and novel work that deserves publication in Nature Communications. However, there are still a couple of issues needed to be addressed and it may be published after major revision. The comments are as follows:

1. The bipolar photocurrent in photodetector has been reported. They need to describe and cite the reported work in the introduction section as "Yang et al. for the first time, reported the wavelength-induced dual-polarity photocurrent in ZnO nanowires based p-n heterojunctions [iScience, 2018, 1, 16-23], which is attributed to the consequence of the competition between photovoltaic effect of ZnO nanowires based p-n heterojunctions and photothermoelectric effect of the p-type thin film [Nano Energy, 2020, 68, 104312; Cell Reports Physical Science, 2021, 2, 100418]."
2. They need to provide the evidence for "the Vis-Perov layer was intrinsic (i-type), and the NIR-Perov layer was the lightly doped p-type".
3. The authors should add more descriptions about the role of PCBM layer. Contrast tests of photodetectors with/without a PCBM layer need to be supplied.
4. More characterizations about the SPPD should be provided, including specific detectivity and photoconductive gain (see e.g. Figure 1b of Advanced Materials 2017, 29, 1703694; Figure 4e Nano Energy 2017, 40, 352-359).
5. They need to illustrate the reason why the energy bands are inclined across the whole Vis-Perov layer and NIR-Perov layer (Fig. 2e).
6. Light intensity of 625 nm light in Fig. 3d and light intensity of 940 nm light in Fig. 3e should be given.

Reviewer #2 (Remarks to the Author):

In this manuscript, Kim et al. reported a self-powered perovskite photodetector (SPPD) with a back-to-back multilayer structure, which achieves optoelectronic logic gates beyond "AND" and "OR" with only one single device, enabling a possible improved integration for the future integrated circuit. After that, the authors demonstrated an 8×8 logic array for potential realistic applications. Overall, this short communication tells an interesting story, and the designed SPPD architecture is novel. The figures are well prepared. I think this work reaches the high standard of Nature Communications and should be published after some revisions. Below are my comments:

1. First and foremost, I think the proposed architecture may have an intrinsic limitation for wide spectrum response. The logic functions in Figures 3 & 4 were realized by using monochromatic 940 nm NIR and 625 nm visible light. In the multilayer structure, both the NIR and visible light were introduced from the bottom ITO side. What would be the result if different wavelengths were used? In figure 2, 530 nm was used for the visible light. Why was 625 nm used for later experiments?
2. In figure 1b, only the elemental mapping images for Sn, I, and Pb, without O and Au, which is in discrepancy with the figure caption. This confusion needs to be clarified.
3. I believe that the perovskite film thickness is critical for the device operation since it determines the light absorption and transmission. How was it optimized?
4. In page 3, for the statement, "The superior optoelectronic properties of organometal halide perovskite have been demonstrated in many studies 27-29", I think more research works on perovskite optoelectronics should be cited. An original work on the broadband perovskite photodetector is ACS Applied Materials & Interfaces 9, 37832-37838 (2017), and a more recent example of taking advantage of optoelectronic properties of perovskites is Applied Physics Reviews 7, 031401 (2020). There are certainly other notable examples the authors can cite and enrich the discussion,

5. What is the reason behind using a device structure for an array (figure 4a) from that for a single logic gate device (figure 1a)? The authors should provide more details.

6. The authors may need to tune down the statement of "ultrafast photoresponse". In figure S7, the authors used an oscilloscope and got a rise/decay time in ms level, and Table S1 compared the speed with other typical reports. However, most of the PDs in Table S1 adopted a planar structure, which has a much slower response speed compared to the vertical structure due to a much wider channel length. On the other hand, many vertical perovskite-based PDs have a speed at μs level or faster, such as one typical example: <https://doi.org/10.1038/ncomms6404>, among many other similar works.

Reviewer #3 (Remarks to the Author):

In this manuscript, the authors demonstrated optoelectronic logic gates based on the bipolar spectral photoresponse characteristics of a perovskite photodiode. The photocurrent polarity of the photodetector was changed using the visible and near-infrared light, and the Boolean output state of "1" or "0" was determined by the positive or negative output photocurrent, respectively. Five representative logic gates ("AND", "OR", "NAND", "NOR", and "NOT") were successfully demonstrated by controlling the photocurrent polarity. In addition, 8x8 device arrays exhibited the high-accuracy operations in the five logic gates.

It was impressive to demonstrate various logic circuits in one device with a unique device configuration. The experimental data were also well-organized. However, although the authors claimed that the photodiodes and logic gates would be useful for an artificial intelligence technology, they did not show the device characteristics related to the artificial intelligence, such as computation capability. In addition, the performance of some devices seems to be overstated, and I think such expressions should be toned down based on the actual measured data. I suggest the following revision of the manuscript before publication.

1. In the introduction, the authors mentioned that the explosive demand for artificial intelligence and big-data has sparked research interest in new logic gate systems. However, to claim that this work is a technology for artificial intelligence, it will be necessary to demonstrate the computing capability using 8x8 device arrays fabricated in Fig. 4. If not, please remove the overstated expressions about artificial intelligence.

2. The authors demonstrated the ultrafast/sensitive photoresponse and the substantial on-off ratio in the back-to-back self-powered perovskite photodetector (SPPD). However, in Supplementary Fig. 7, the operating speed of milliseconds looks quite slow. Please specify the reason.

3. I strongly suggest that the authors should demonstrate the device array with a reduced pixel size of less than a few microns (Fig. 4), for the comparison with the current technology.

4. Perovskite-based devices typically have the weakness on retention and endurance. For the practical implementation, it will be important to show the reliable and repeated operation for a long period.

5. It is not clear why the optoelectronic logic gate platform can suggest an integrated chip for optical computing, optical communication, logic memory units, and photonic quantum computation (in the conclusion section). More detailed explanation is necessary even though the authors simply mentioned the future plan.

Reviewer #1

In this work, the authors reported multifunctional logic gates which was constructed based on bipolar photoresponse from single photodetector. This is an interesting and novel work that deserves publication in Nature Communications. However, there are still a couple of issues needed to be addressed and it may be published after major revision. The comments are as follows:

Q1) The bipolar photocurrent in photodetector has been reported. They need to describe and cite the reported work in the introduction section as "Yang et al. for the first time, reported the wavelength-induced dual-polarity photocurrent in ZnO nanowires based p-n heterojunctions [iScience, 2018, 1, 16-23], which is attributed to the consequence of the competition between photovoltaic effect of ZnO nanowires based p-n heterojunctions and photothermoelectric effect of the p-type thin film [Nano Energy, 2020, 68, 104312; Cell Reports Physical Science, 2021, 2, 100418]."

[Answer]

We appreciate this comment. We gained crucial knowledge and clues from the references suggested by the reviewer #1.

As you commented, Yang et al., for the first time, reported the wavelength-induced dual-polarity property from a *p-n* heterojunction diode composed of *n*-type ZnO nanowires and *p*-type thermoelectric thin film (SnS or Sb₂Se₃)^{R1-R3}. According to the report, the dual-polarity occurs due to the coexistence of two photocurrent generation mechanisms: photovoltaic (PV) effect occurring by band-to-band transition and photo-thermoelectric (PTE) effect driven by photo-induced temperature-gradient^{R1-R3}. In their diodes, short-wavelength lights are mainly absorbed by the thermoelectric Sb₂Se₃ film. In contrast, long-wavelength lights are absorbed at the vicinity of the *p-n* junction. This finding indicates that the direction and amount of photocurrent can be controlled depending on the wavelength and power density.

Therefore, we decided to importantly describe all the suggested references in the revised manuscript, as we think they would be very beneficial in solidifying our paper's theoretical basis. Thank you again for the great suggestion.

[Action]

We cited the main contents of the suggested references on **page 3** of the revised manuscript as follows:

Manuscript (page 3, paragraph 1): For instance, Yang *et al.* reported the wavelength-induced dual-polarity phenomenon from a *p-n* heterojunction diode composed of *n*-type ZnO nanowires and *p*-type thermoelectric thin film (SnS or Sb₂Se₃)¹⁹⁻²¹.

19. Quyang, B. et al. Dual-polarity output response-based photoelectric devices. *Cell Rep. Phys. Sci.* **2**, 100418 (2021).
20. Quyang, B. et al. Dual-polarity response in self-powered ZnO NWs/Sb₂Se₃ film heterojunction photodetector array for optical communication. *Nano Energy* **68**, 104312 (2020).
21. Quyang, B. et al. Photocurrent polarity controlled by light wavelength in self-powered ZnO nanowires/SnS photodetector system. *iScience* **1**, 16-23 (2018).

Q2) They need to provide the evidence for “the Vis-Perov layer was intrinsic (i-type), and the NIR-Perov layer was the lightly doped p-type”.

[Answer]

In the original manuscript, we provided the energy band diagram of the $p^+ - i - n - p - p^+$ structure configured with the results of UPS and UV-Vis measurements (Fig. 1d). The difference between the Fermi-level and the mid-gap state is 0.04 for the Vis-Perov (MAPbI₃) and 0.1 eV for the NIR-Perov (MA_{0.5}FA_{0.5}Pb_{0.4}Sn_{0.6}I₃). Therefore, in the original manuscript, we described that the NIR-Perov is a lightly doped p -type.

We agree that another strong evidence for the doping level is required. For this purpose, we conducted Hall measurement for ten samples of each perovskite film (Vis-Perov and NIR-Perov). Average Hall values were calculated as shown in the Table R1 below.

Table R1. Comparison of the Hall values (carrier mobility and concentration) of Pb and Sn-based perovskites.

Materials	Mobility (cm ² V ⁻¹ s ⁻¹)	Carrier concentration (cm ⁻³)	Reference
MA _{0.5} FA _{0.5} Pb _{0.4} Sn _{0.6} I ₃ (NIR-Perov)	27	2.4×10 ¹⁵	This work
MAPbI ₃ (Vis-Perov)	-	~10 ¹²	This work
FASnI ₃	5–40	8×10 ¹⁵ –7×10 ¹⁶	R4
FASnI ₃	19	2.4×10 ¹⁶	R5
MAPbI ₃	15.9	1.02×10 ¹²	R6
MAPbI ₃	58.79	3.375×10 ¹¹	R7

From the Hall measurements, we confirmed that the NIR-Perov has a mobility of 27 cm²V⁻¹s⁻¹ and a hole concentration of 2.4×10¹⁵ cm⁻³, which are very similar to the reported values (p -type Sn-based perovskites)^{R4,R5}. Considering the location of Fermi-level (Fig. 1d) and the hole concentration level of ~10¹⁵ cm⁻³, the NIR-Perov is a lightly doped p -type semiconductor.

The carrier concentration of Vis-Perov was measured as ~10¹² cm⁻³, but the mobility was

not reliably measured due to its low carrier concentration. We found in literature that intrinsic MAPbI₃ thin films had a concentration of 10¹¹–10¹² cm⁻³ R6-R7. Therefore, the Vis-Perov film is regarded as an intrinsic semiconductor.

[Action]

We modified a sentence on **page 6** of the revised manuscript and summarized Hall measurement results for the Vis-Perov and NIR-Perov films in **Supplementary Table 1** (Table R1) in the revised supplementary information.

Manuscript (page 6, paragraph 2): The Vis-Perov layer was intrinsic (*i*-type), and the NIR-Perov layer was lightly doped *p*-type, which were additionally supported by Hall measurements (Supplementary Table 1).

Q3) The authors should add more descriptions about the role of PCBM layer. Contrast tests of photodetectors with/without a PCBM layer need to be supplied.

[Answer]

The PCBM layer serves as an intermediate *n*-type semiconductor playing a key role for realizing the bipolar photoresponse of $p^+i-n-p-p^+$ device. To confirm the role of PCBM layer, we fabricated a PCBM-less device (the inset of Fig. R1a). Both NIR and visible lights drove photocurrents in the same direction (Vis-Perov \rightarrow NIR-Perov), as shown in Fig. R1a. The photocurrent was markedly suppressed to $< 0.1 \mu\text{A}$, 20 times smaller than that of the PCBM-interposed device.

The unidirectional photocurrent generation in the PCBM-less device can be explicated based on the energy band diagram depicted in Fig. R1b. The large valence band offset (ΔE_v) between the two perovskites can influence hole transport. The deep HOMO level of the Vis-Perov, located at 0.64 eV lower than that of the NIR-Perov, facilitates hole transport from Vis-Perov to NIR-Perov, simultaneously hindering the reverse flow. Therefore, photo-generated holes are allowed to flow in only one direction regardless of incident wavelength.

In contrast, the PCBM with a high electron density can block the hole transport, as depicted in Fig. 1d and 2e. Thus, only photo-generated electrons can move through the PCBM layer and be accumulated in the counter perovskite. Therefore, the photocurrent direction can be readily controlled by changing the incident light wavelength when the PCBM is interposed.

Fig. R1 Optoelectronic properties of the PCBM-less PD. a Steady-state currents of the PCBM-less device under 625 and 940 nm irradiation (1 mW cm^{-2}). **b** Energy band diagram of the PCBM-less PD.

[Action]

We added a sentence for the role of PCBM layer on **page 9** of the revised manuscript. The optoelectronic properties of the PCBM-less device can be found in **Supplementary Fig. 15**.

Manuscript (page 9, paragraph 2): It is noteworthy that when the PCBM layer was not inserted, the photocurrent flowed unidirectionally from Vis-Perov to NIR-Perov as shown in Supplementary Fig. 15.

Q4) More characterizations about the SPPD should be provided, including specific detectivity and photoconductive gain (see e.g. Figure 1b of *Advanced Materials* 2017, 29, 1703694; Figure 4e *Nano Energy* 2017, 40, 352-359).

[Answer]

We calculated photoconductive gain (G_λ) and specific detectivity (D_λ) using the following formulas adopted in the suggested papers^{R9,R10}

$$G_\lambda = \frac{R_\lambda \cdot h \cdot c}{q \cdot \lambda} \quad (R1)$$

$$D_\lambda = \frac{R_\lambda}{\sqrt{2 \cdot q \cdot I_d \cdot A}} \quad (R2)$$

,where R_λ is a responsivity of photodetector, recorded under a wavelength of light of λ , I_d and A are a dark current and active area of the device, respectively; h , c , and q are physical constants representing Planck constant, speed of light, and unit charge, respectively. Note that we employed specific detectivity (D_λ), with the assumption that a shot noise dominantly determines a noise current of photodetector.

The calculated photoconductive gains under 530 nm, 625 nm, and 940 nm light illumination were 0.02, 0.007 and 0.008, respectively. The corresponding detectivities were calculated as 1.89 , 0.76 , and 1.27×10^{11} Jones, respectively.

Fig. R2 a Photoconductive gain and **b** specific detectivity of the SPPD under wavelength from 400 to 1000 nm.

[Action]

We added new sentences for the calculated G_λ and D_λ values on **page 7** of the revised manuscript. The detailed description can be found in **Supplementary Fig. 9**.

Manuscript (page 7, paragraph 3): Based on the spectral responsivity results, photoconductive gain (G_λ) and detectivity (D_λ) values were calculated as shown in Supplementary Fig. 9. The G_λ values were 0.02 and 0.008 for 530 and 940 nm light illumination, respectively. The corresponding D_λ values were calculated as 1.89 and 1.27×10^{11} Jones, respectively.

Q5) They need to illustrate the reason why the energy bands are inclined across the whole Vis-Perov layer and NIR-Perov layer (Fig. 2e).

[Answer]

A built-in potential caused by a Fermi-level difference leads to band-bending at material interfaces, as shown in Fig. R3a^{R11-R13}. Herein, we may ignore the band-bending at the interfaces between the perovskites and hole transport layers (PEDOT:PSS or Spiro-OMeTAD) because of the minute Fermi-level difference. Then, we can simply consider the heterojunction between the PCBM and perovskite regarding on this matter. The built-in potential across the perovskite can be estimated as

$$V(x) = -V_i + \frac{qN_a}{2\epsilon_{\text{Perov}}}(x - x_p)^2 \quad (\text{R3})$$

,where V_i is the Fermi-level difference between the PCBM and perovskite; ϵ_{Perov} and N_a are the dielectric constant and doping concentration of perovskite, respectively; q is the unit charge; x is the perovskite depth from the interface^{R11-R13}; and x_p is the thickness of depletion zone (energy-bending region) which is defined as

$$x_p = \left[\frac{2V_i \epsilon_{\text{PCBM}} \epsilon_{\text{Perov}} N_d}{qN_a (\epsilon_{\text{PCBM}} N_d + \epsilon_{\text{Perov}} N_a)} \right]^{1/2} \quad (\text{R4})$$

,where ϵ_{PCBM} and N_d are the dielectric constant and doping concentration of PCBM layer, respectively. We measured the hole concentration of NIR-Perov ($2.40 \times 10^{15} \text{ cm}^{-3}$) and Vis-Perov ($1.00 \times 10^{12} \text{ cm}^{-3}$) through Hall measurement, as described in question #2. The other parameters (V_i and ϵ) were investigated through UPS (Supplementary Fig. 5) and literature survey^{R6,R7,R14}. All information on the perovskites and PCBM are summarized in Fig. R3b. From the formula above, we confirmed that the theoretical depletion width (x_p) of Vis-Perov is above $10 \mu\text{m}$ which is much thicker than the actual thickness ($\sim 200 \text{ nm}$). It implies that the film was fully depleted, rendering the inclined energy band through the entire film. This estimation can be well accepted considering the intrinsic Vis-Perov. The x_p of NIR-Perov was calculated as $\sim 220 \text{ nm}$, which is more than two-thirds of the NIR-Perov thickness. Thus, the energy band of NIR-Perov is also expected to be bent across the film.

Fig. R3 Energy band bending of perovskite in the SPPD. a Schematic of band bending of a perovskite. **b** Inherent parameters for each layers in the SPPD: V_i is the relative Fermi-level difference with the PCBM; N_a (N_d) is the hole (electron) concentration; ϵ is the dielectric constant (ϵ_0 is the vacuum permittivity); and x_p is the theoretical depletion width.

[Action]

We added a sentence on **page 9** of the revised manuscript to provide information on the built-in potential and energy band-bending. The above discussion (for Fig. R3) was added in **Supplementary Fig. 14**.

Manuscript (page 9, paragraph 2): More details on the built-in potential and energy band-bending are explicated in Supplementary Fig. 14.

Q6) They need Light intensity of 625 nm light in Fig. 3d and light intensity of 940 nm light in Fig. 3e should be given.

[Action]

We made actions in Fig. 3d ~ 3g to improve the readability per request.

Revised Fig. 3d ~ g

[References]

- R1. Quyang, B. et al. Dual-polarity output response-based photoelectric devices. *Cell Rep. Phys. Sci.* **2**, 100418 (2021).
- R2. Quyang, B. et al. Dual-polarity response in self-powered ZnO NWs/Sb₂Se₃ film heterojunction photodetector array for optical communication. *Nano Energy* **68**, 104312 (2020).
- R3. Quyang, B. et al. Photocurrent polarity controlled by light wavelength in self-powered ZnO nanowires/SnS photodetector system. *iScience* **1**, 16-23 (2018).
- R4. Tang, G. et al. Synergistic effects of the zinc acetate additive on the zinc acetate additive on the performance enhancement of Sn-based perovskite solar cells. *Mater. Chem. Front.* **5**, 1995-2000 (2021).
- R5. Liu, C.-K. et al. Sn-based perovskite for highly sensitive photodetectors. *Adv. Sci.* **6**, 1900751 (2019).
- R6. Yang, Y. et al. Effect of doping of NaI monovalent cation halide on the structural, morphological, optical and optoelectronic properties of MAPbI₃ perovskite. *J. Mater. Sci: Mater. Electron.* **29**, 205-210 (2018).
- R7. Li, Y. et al. Highly conductive P-type MAPbI₃ films and crystals via sodium doping. *Font. Chem.* **8**, 754 (2020).
- R8. Shi, D. et al. Low trap-state density and long carrier diffusion in organolead trihalide perovskite single crystals. *Science* **347**, 519-522 (2015).
- R9. Ma, N. et al. Photovoltaic–pyroelectric coupled effect induced electricity for self-powered photodetector system. *Adv. Mater.* **29**, 1703694 (2017).
- R10. Ma, N & Yang, Y. Enhanced self-powered UV photoresponse of ferroelectric BaTiO₃ materials by pyroelectric effect. *Nano Energy* **40**, 352-359 (2017).
- R11. Guerrero, A. et al. Electrical field profile and doping in planar lead halide perovskite solar cells. *Appl. Phys. Lett.* **105**, 133902 (2014).
- R12. Crovetto, A. et al. How the relative permittivity of solar cell materials influences solar cell performance. *Solar Energy* **149**, 145-150 (2017).
- R13. Moeini, I. et al. Modeling the time-dependent characteristics of perovskite solar cells. *Solar Energy* **170**, 969-973 (2018).
- R14. Maibach, J. et al. The band energy diagram of PCBM–DH6T bulk heterojunction solar cells: synchrotron-induced photoelectron spectroscopy on solution processed DH6T:PCBM blends and in situ prepared PCBM/DH6T interfaces. *J. Mater. Chem. C* **1**, 7635-7642 (2013).

Reviewer #2

In this manuscript, Kim et al. reported a self-powered perovskite photodetector (SPPD) with a back-to-back multilayer structure, which achieves optoelectronic logic gates beyond “AND” and “OR” with only one single device, enabling a possible improved integration for the future integrated circuit. After that, the authors demonstrated an 8×8 logic array for potential realistic applications. Overall, this short communication tells an interesting story, and the designed SPPD architecture is novel. The figures are well prepared. I think this work reaches the high standard of Nature Communications and should be published after some revisions. Below are my comments:

Q1) First and foremost, I think the proposed architecture may have an intrinsic limitation for wide spectrum response. The logic functions in Figures 3 & 4 were realized by using monochromic 940 nm NIR and 625 nm visible light. In the multilayer structure, both the NIR and visible light were introduced from the bottom ITO side. What would be the result if different wavelengths were used? In figure 2, 530 nm was used for the visible light. Why was 625 nm used for later experiments?

[Answer]

Thank you for this comment. We agree that the data using different wavelengths in Fig. 2 and 3 may confuse readers.

Delicate tuning of the logic functions via the light intensity modulation for the opposite photocurrent-offset can be most efficient when the combination of two wavelengths having a similar photoconductive gain is applied as the input and modulator. As shown in Supplementary Fig. 9a, the photoconductive gain strongly depends on the wavelength of light. The photoconductive gain for the 625 nm light was 0.007, almost similar to that (0.008) for the 940 nm light. The photoconductive gain for 530 nm light is 0.02, much larger than that for the 625 nm. It means that the delicate tuning of the logic functions becomes much difficult. In this case, the irradiance of 530 nm, equivalent to a third of that of the 625 nm, should be used to achieve optimal logic outputs. Then, the five logics were faithfully implemented in the same SPPD device under 530 and 940 nm illumination (Fig. R4).

Fig. R4 Logic gate operation of the $50 \times 50 \mu\text{m}^2$ pixel array using a pair of 530 and 940 nm light sources. Three-dimensional bar charts for all the outputs ("OR", "AND", "NAND", "NOR", and "NOT") obtained from the 16 pixels in SPPD-OELG array.

In addition, we investigated I - V characteristics of the SPPD using the 625 nm light. As expected, the sign of V_{OC} (Fig. R5a) was still negative. It means that the red (625 nm) light can replace the green (530 nm) light in our logic device. However, as discussed above, the irradiance should be considered for the efficient current-offset.

Fig. R5 Optoelectronic properties of the SPPD. **a**, **b** I - V curves and steady-state currents at zero bias under dark and irradiation conditions (625 and 940 nm at 1 mW cm^{-2}).

[Action]

We added additional sentences on **page 10 and 12** of the revised manuscript to explain the use of 625 nm for logic operations rather than 530 nm. The optoelectronic properties using the 625 nm light (Fig. R5) were added in **Supplementary Fig. 16**, and the logic gate operation using the combination of 530 and 940 nm lights (Fig. R4) were discussed in **Supplementary Fig. 17**.

Manuscript (page 10, paragraph 2): Delicate tuning of logic functions via the light intensity modulation for the opposite photocurrent-offset can be most efficient when the two light wavelengths have similar photoconductive gains. As the 625 nm light revealed a well matched photoconductive gain (0.007) with that (0.008) in case of the 940 nm light (Supplementary Fig. 9a), hereafter, we utilized these two light sources for the following logic gate operations. Prior to the use of 625 nm, we confirmed that the 625 nm irradiation generated the opposite photocurrents compared to the 940 nm case (Supplementary Fig. 16).

Manuscript (page 12, paragraph 2): In addition, we also confirmed logic gate operations using a pair of 530 and 940 nm light sources after precisely adjusting the irradiance of 530 nm light (Supplementary Fig. 17).

Q2) In figure 1b, only the elemental mapping images for Sn, I, and Pb, without O and Au, which is in discrepancy with the figure caption. This confusion needs to be clarified.

[Action]

We are sorry for the confusion. O and Au mapping images were added in revised Fig. 1b.

Revised Fig. 1b

Q3) I believe that the perovskite film thickness is critical for the device operation since it determines the light absorption and transmission. How was it optimized?

[Answer]

Before device fabrication, we simulated theoretical photoconductive gain (G_λ) by varying the thickness of perovskite films to explore an optimized condition. As shown in Fig. R6a, the color and tone represent the gain ratio (G_{940}/G_{625}), which is distributed on the plane as a function of perovskite film thickness ranging from 50 to 400 nm. The gain ratio can be obtained using the formula below:

$$\frac{\int_{T_{NIR-Perov}} \frac{hc}{925} C_R(940, x) dx}{\int_{T_{Vis-Perov}} \frac{hc}{625} C_R(625, x) dx - \int_{T_{NIR-Perov}} \frac{hc}{625} C_R(625, x) dx} \quad (R5)$$

,where h and c are Planck constant and speed of light, respectively; $T_{NIR-Perov}$ and $T_{Vis-Perov}$ represent the thickness of NIR-Perov and Vis-Perov, respectively; $C_R(\lambda, x)$ is the charge generation rate at a certain depth position x of device under illumination of λ nm light source with unit irradiance (1 mW cm^{-2}).

Our logic gate works based on the offset of opposite photocurrents generated under the two light sources. Therefore, the logic gate operation becomes more efficient and controllable when the gain ratio becomes 1. The thickness condition (the combination of 300 nm NIR-Perov and 200 nm Vis-Perov) used for our logic gate devices ('Device B') in the original manuscript locates within the optimized green zone (diamond symbol) at which the gain ratio is ~ 1 , as shown in Fig. R6a.

A validation test was performed by comparing the photoresponses of three devices (denoted 'Device A', 'Device B' and 'Device C' in Fig. R6a) and their photoconductive gain ratios were investigated under 625 and 940 nm light illumination (Fig. R6b and c). Under the two light sources, the photoresponses of both 'Device A' and 'Device C', having a perovskite layer thickness less than 150 nm, are not as balanced as that of 'Device B', which is not good for the logic gate operation (Fig. 6c). The combination of 300 nm NIR-Perov / 200 nm Vis-Perov has the gain ratio of 1, optimal for the efficient logic gate operation.

Fig. R6 Investigation on optimized perovskite film thickness. **a** Simulation of photoconductive gain ratio (G_{940}/G_{625}) with different perovskite thicknesses. **b** Empirical and theoretical gain ratio values of the three devices; marked in **a**. **c** Steady-state currents of the device A, B, and C under 625 and 940 nm illumination.

[Action]

We added an additional sentence on **page 5** of the revised manuscript to show how we optimized the perovskite thickness for the efficient logic gate operation. The simulation results discussed above were added in **Supplementary Fig. 2**.

Manuscript (page 5, paragraph 1): The thickness of each perovskite layer (300 nm NIR-Perov and 200 nm Vis-Perov) was decided via a theoretical simulation, aiming a photoconductive gain ratio (G_{940}/G_{625}) of 1 for the efficient current offset, which is beneficial for the tuning of logic gate operation (Supplementary Fig. 2).

Q4) In page 3, for the statement, “The superior optoelectronic properties of organometal halide perovskite have been demonstrated in many studies 27-29”, I think more research works on perovskite optoelectronics should be cited. An original work on the broadband perovskite photodetector is ACS Applied Materials & Interfaces 9, 37832-37838 (2017), and a more recent example of taking advantage of optoelectronic properties of perovskites is Applied Physics Reviews 7, 031401 (2020). There are certainly other notable examples the authors can cite and enrich the discussion.

[Answer]

We delight in citing the articles that provide information on the unique optoelectronic properties of perovskites.

Roqan et al. contrived a broadband perovskite (MAPbI₃) photodetector, detectable from UV to NIR range (250–1350 nm) by exploiting Gd-doped ZnO nanorods^{R15}. The device responded to UV and visible lights based on the band-to-band transition mechanism. Moreover, they discovered that Fano-like intraband transition occurs under the wavelength of 1350 nm, bringing about a significant photoresponse.

Wu et al. have exploited the optoelectronic properties of perovskites to realize an artificial iconic photo-memory^{R16}. They fabricated a cross-bar type array of which a pixel consists of a two-terminal device configuration in the form of ITO/Perovskite/Au/Perovskite/Ag. They demonstrated the capability to capture and memorize an image in the form of nonvolatile data.

[Action]

We cited the suggested references on **page 3** of the revised manuscript as follows:

Manuscript (page 3, paragraph 2): The superior optoelectronic properties of organometal halide perovskite have been demonstrated in many studies³⁰⁻³⁴.

33. Alwadai, N. et al. High-performance ultraviolet-to-infrared broadband perovskite photodetectors achieved via inter-/intraband transitions. *ACS Appl. Mater. Interfaces* **9**, 37832-37838 (2017).
34. Guan, X. et al. A monolithic artificial iconic memory based on highly stable perovskite-metal multilayers. *Appl. Phys. Rev.* **7**, 031401 (2020)

Q5) What is the reason behind using a device structure for an array (figure 4a) from that for a single logic gate device (figure 1a)? The authors should provide more details.

[Answer]

We appreciate this comment as it can significantly help to enrich the manuscript. The single device (from Fig. 1 to 3) was made to investigate device characterization and working principle for the logic gate operation. We desired to make a more practical platform being urged in real-world applications for data processing. In this perspective, it is imperative to construct an integrated logic gate platform and investigate its accuracy to ensure the practical feasibility for a chip-level system. Herein, we succeeded in constructing an 8×8 crossbar-type SPPD array in which five different logics were faithfully implementable by individual pixel in an array at a 100% precision. The integrated SPPD-OELG array exhibits a superior reproducibility for tunable logic gate operations.

[Action]

We made modification on **page 13** of the revised manuscript as follows:

Manuscript (page 13, paragraph 1): For real-world applications, it is imperative to construct integrated SPPD-OELG platform and investigate its accuracy to ensure the practical chip-level feasibility.

Q6) The authors may need to tune down the statement of “ultrafast photoresponse”. In figure S7, the authors used an oscilloscope and got a rise/decay time in ms level, and Table S1 compared the speed with other typical reports. However, most of the PDs in Table S1 adopted a planar structure, which has a much slower response speed compared to the vertical structure due to a much wider channel length. On the other hand, many vertical perovskite-based PDs have a speed at μs level or faster, such as one typical example: <https://doi.org/10.1038/ncomms6404>, among many other similar works.

[Answer]

We agree with the reviewer's opinion. In the revised manuscript, we toned down the over-expression of 'ultrafast'.

As shown in Fig. R7, we confirmed that the rise/decay times of perovskite PD could be shortened to tens of microseconds by reducing the active area by 100 times (from 4.64 mm^2 to 0.04 mm^2), even though there is still an apparent gap compared to the state-of-art levels (e.g., hundreds of ns)^{R17-R19}.

We believe that it can be enhanced by sophisticated pixel-downsizing. However, at the current lithographic and vacuum technology, it is difficult to downsize to a few micron size without perovskite degradation. In other way, producing a high-crystalline perovskite film to depress charge-trapping can be a realistic approach to enhance the response speed^{R17-R19}.

Thank you again for this constructive comment.

Fig. R7 Operating speed of the SPPD with an active area of 0.04 mm^2 . a, b Rise and decay times under 530 nm light illumination at the irradiance of 1 mW cm^{-2} . c, d Rise and decay times under 940 nm light illumination at the same irradiance. The rise time was defined as the time required for the current to reach 90 % of the saturation level under light illumination, while the decay time was the delayed time required to fall to 10 % of the saturation level.

[Action]

We changed with new results on **page 7** of the revised manuscript. Details on the rise/decay times of the SPPD with active area of 0.04 mm^2 were described in **Supplementary Fig. 8**. The reference lists in **Supplementary Table S2** were edited with only vertical perovskite-based photodetectors.

Manuscript (page 7, paragraph 2): The rise/decay times of SPPD with an active area of 0.04 mm^2 were $62/680 \text{ } \mu\text{s}$ for 530 nm and $40/72 \text{ } \mu\text{s}$ for 940 nm (Supplementary Fig. 8). Even though there is still an apparent gap compared to the state-of-art level (e.g., hundreds of ns, Supplementary Table 2), it can be enhanced using a high-crystalline perovskite film with further downsizing the active area.

[References]

- R15. Alwadai, N. et al. High-performance ultraviolet-to-infrared broadband perovskite photodetectors achieved via inter-/intrapband transitions. *ACS Appl. Mater. Interfaces* **9**, 37832-37838 (2017).
- R16. Guan, X. et al. A monolithic artificial iconic memory based on highly stable perovskite-metal multilayers. *Appl. Phys. Rev.* **7**, 031401 (2020).
- R17. Dou, L et al. Solution-processed hybrid perovskite photodetectors with high detectivity. *Nat. Commun.* **5**, 5404 (2014).
- R18. Chen, Z. et al. Solution-processed visible-blind ultraviolet photodetectors with nanosecond response time and high detectivity. *Adv. Opt. Mater.* **7**, 1900506 (2019).
- R19. Shen, L. et al. A self-powered, sub-nanosecond-response solution processed hybrid perovskite photodetector for time resolved photoluminescence-lifetime detection. *Adv. Mater.* **28**, 10794-10800 (2016).

Reviewer #3

In this manuscript, the authors demonstrated optoelectronic logic gates based on the bipolar spectral photoresponse characteristics of a perovskite photodiode. The photocurrent polarity of the photodetector was changed using the visible and near-infrared light, and the Boolean output state of "1" or "0" was determined by the positive or negative output photocurrent, respectively. Five representative logic gates ("AND", "OR", "NAND", "NOR", and "NOT") were successfully demonstrated by controlling the photocurrent polarity. In addition, 8x8 device arrays exhibited the high-accuracy operations in the five logic gates.

It was impressive to demonstrate various logic circuits in one device with a unique device configuration. The experimental data were also well-organized. However, although the authors claimed that the photodiodes and logic gates would be useful for an artificial intelligence technology, they did not show the device characteristics related to the artificial intelligence, such as computation capability. In addition, the performance of some devices seems to be overstated, and I think such expressions should be toned down based on the actual measured data. I suggest the following revision of the manuscript before publication.

Q1) In the introduction, the authors mentioned that the explosive demand for artificial intelligence and big-data has sparked research interest in new logic gate systems. However, to claim that this work is a technology for artificial intelligence, it will be necessary to demonstrate the computing capability using 8x8 device arrays fabricated in Fig. 4. If not, please remove the overstated expressions about artificial intelligence.

[Answer]

Humbly accepting the reviewer's comment, we deleted the overstated terminologies and descriptions to prevent confusing potential readers.

[Action]

We modified the overstated expression on **page 1 and 2** of the revised manuscript and added paragraphs as follows:

Manuscript (page 1, paragraph 1): The explosive demand for a wide range of data processing has sparked interest towards a new logic gate platform as the existing electronic logic gates face limitations in accurate and fast computing.

(In the original manuscript) The explosive demand for artificial intelligence and big-data has sparked interest towards a new logic gate platform as the existing electronic or all-optical logic gates face limitations in accurate and fast data processing.

Manuscript (page 2, paragraph 1): Optoelectronic logic gates (OELGs) are receiving significant attention as crucial building block of future integrated circuits for accurate and fast data processing¹⁻⁴. Existing circuits or processors based on electronic logic gates would face limitations in computing extensive data sets which are expected to markedly increase in the fourth industrial revolution age, due to performance shortfalls in switching, operation, computing, and decision making/regenerating^{5,6}. Thus, developing an innovative logic gate platform that can implement faster computation with less power consumption is imperative to fulfill upcoming new computing trends.

(In the original manuscript) Optoelectronic logic gates (OELGs) are a crucial building block of the future integrated circuits because the fourth industrial revolution requires a novel computing system that enables unprecedentedly accurate and fast data processing¹⁻³. The explosive demand for artificial intelligence and big-data has sparked research interest in new logic gate systems⁴. Existing circuits or processors based on electronic logic gates face limitations in the sophisticated analysis and rapid extraction of information due to performance shortfalls in switching, computing, and decision making/regenerating^{5,6}. Thus, it is imperative to develop an innovative logic gate platform that can be applied to efficiently perceive hidden values, such as patterns, trends, and associations from extensive data sets that seem meaningless at first glance.

Q2) The authors demonstrated the ultrafast/sensitive photoresponse and the substantial on-off ratio in the back-to-back self-powered perovskite photodetector (SPPD). However, in Supplementary Fig. 7, the operating speed of milliseconds looks quite slow. Please specify the reason.

[Answer]

We agree with the reviewer's opinion. In the revised manuscript, we toned down the over-expression of 'ultrafast'.

As shown in Fig. R7, we confirmed that the rise/decay times of perovskite PD could be shortened to tens of microseconds by reducing the active area by 100 times (from 4.64 mm² to 0.04 mm²), even though there is still an apparent gap compared to the state-of-art levels (e.g., hundreds of ns)^{R17-R19}.

We believe that it can be enhanced with by sophisticated pixel-downsizing. However, at the current lithographic and vacuum technology, it is difficult to downsize to a few micron size without perovskite degradation. In other way, producing a high-crystalline perovskite film to depress charge-trapping can be a realistic approach to enhance the response speed^{R17-R19}.

Thank you again for this constructive comment.

Fig. R7 Operating speed of the SPPD with an active area of 0.04 mm². **a, b** Rise and decay times under 530 nm light illumination at the irradiance of 1 mW cm⁻². **c, d** Rise and decay times under 940 nm light illumination at the same irradiance. The rise time was defined as the time required for the current to reach 90 % of the saturation level under light illumination, while the decay time was the delayed time required to fall to 10 % of the saturation level.

[Action]

We changed with new results on **page 7** of the revised manuscript. Details on the rise/decay times of the SPPD with active area of 0.04 mm² were described in **Supplementary Fig. 8**. The reference lists in **Supplementary Table S2** were edited with only vertical perovskite-based photodetectors.

Manuscript (page 7, paragraph 2): The rise/decay times of SPPD with an active area of 0.04 mm² were 62/680 μs for 530 nm and 40/72 μs for 940 nm (Supplementary Fig. 8). Even though there is still an apparent gap compared to the state-of-art level (e.g., hundreds of ns, Supplementary Table 2), it can be enhanced using a high-crystalline perovskite film with further downsizing the active area.

Q3) I strongly suggest that the authors should demonstrate the device array with a reduced pixel size of less than a few microns (Fig. 4), for the comparison with the current technology.

[Answer]

We strongly agree that a perovskite PD array at submicron scale is critical for practical applications. To fabricate such a sub-micron scaled pattern array, highly resolved photo- or e-beam lithography should be used but, access to those tools and severe degradations in organic layers (perovskite, PCBM, PEDOT:PSS, and Spiro-OMeTAD)^{R20,R21} are problematic during the lithography process.

As per the reviewer's request, we tried our best to reduce the pixel size by using shadow-mask-assisted thermal evaporation at $\sim 50 \mu\text{m}$ width ($200 \mu\text{m}$ in the original manuscript). We appreciate it if the reviewer understands the difficulty of making submicron-scale pixels.

Fig. R8 shows optical microscope images of the 4×4 array at a pixel size of 100×100 and $50 \times 50 \mu\text{m}^2$. Fig. R9 and R10 present the logic gate operations of the 100×100 and $50 \times 50 \mu\text{m}^2$ pixel array, respectively. The photocurrent was reduced with decreasing the active pixel size, but the output polarity was maintained in every pixel, faithfully yielding all the five logic operations.

Fig. R8 Optical microscope image of 4×4 arrays with the pixel size of **a** $100 \times 100 \mu\text{m}^2$ and **b** $50 \times 50 \mu\text{m}^2$, respectively.

Fig. R9 Logic gate operation of the $100 \times 100 \mu\text{m}^2$ pixel array. Three-dimensional bar charts for all the outputs ("OR", "AND", "NAND", "NOR", and "NOT") obtained from the 16 pixels.

Fig. R10 Logic gate operation of the $50 \times 50 \mu\text{m}^2$ pixel array. Three-dimensional bar charts for all the outputs ("OR", "AND", "NAND", "NOR", and "NOT") obtained from the 16 pixels.

[Action]

We added a new sentence regarding the pixel down-sizing effect on **page 13** of the revised manuscript, and the new results (logic gate operations of 100×100 and $50 \times 50 \mu\text{m}^2$ pixel arrays) were added in **Supplementary Fig. 18-20**.

Manuscript (page 13, paragraph 2): Even after down-sizing the pixel to $50 \times 50 \mu\text{m}^2$, the SPPD array succeeded in yielding all the logic gate operations with 100% accuracy. The photocurrent was reduced with decreasing the active pixel size, but the output polarity was maintained in every pixel (Supplementary Fig. 18-20).

Q4) Perovskite-based devices typically have the weakness on retention and endurance. For the practical implementation, it will be important to show the reliable and repeated operation for a long period.

[Answer]

As requested, we conducted the repeatability and long-term stability test of the SPPD under 530 nm (Fig. R11a) and 940 nm (Fig. R11b) light pulses (5s on / 5s off). The photocurrent values are normalized with respect to the photocurrent recorded at the first pulse. Each magnified photocurrent behavior recorded after 4 hrs clearly shows the on-off current behavior without degradation. In this test, the SPPD retained its initial responsivity even after 8 hrs (approximately 3000 pulses) under ambient conditions.

For long-term stability testing, we analyzed a SPPD that had been stored in laboratory for one year under dry-air condition (Fig. R12). The saturated on-current, under 530 and 940 nm lights, was reduced to 90% and 50% of its initial value, respectively. The relatively significant reduction under the NIR light can be attributed to the instability of Sn cations of the NIR-Perov in the air^{R22}.

The relatively severe degradation in the NIR-Perov indicates that the output of logic gate would not be correct or constant any longer. Encapsulating the SPPD array with hydrophobic polymers (Polyisobutylene^{R23}, photocurable fluoropolymers^{R24}, Teflon^{R25}, or Parylene C^{R26}) or compact insulating oxides (Al₂O₃)^{R27-R29} can effectively retard the degradation by preventing the access of oxidizing gas molecules (O₂ or H₂O).

Fig. R11 Repeatability tests of the SPPD. **a, b** Steady-state currents under light pulses (5 s on and 5 s off) at a wavelength of **a** 530 nm and **b** 940 nm. The measurements were conducted for 3000 pulses under ambient conditions, and the right inset figures represent the magnified current behavior after 4 hrs. The current values for each figure were normalized with respect to the photocurrent recorded at the first pulse.

Fig. R12 Long-term stability of the SPPD. **a, b** Steady-state currents under illumination of **a** 530 nm and **b** 940 nm lights after 1 year storage under dry-air condition.

[Action]

We added new sentences regarding the repeatability and long-term stability results on **page 8** of the revised manuscript. Detailed discussion on the new results can be seen in **Supplementary Fig. 10 and 11**.

Manuscript (page 8, paragraph 2): The SPPD retained its initial responsivity for 3000 pulses (5s on / 5s off) under ambient conditions as shown in Supplementary Fig. 10. For long-term stability testing, the SPPD was stored in dry-air condition for one year. The saturated on-current was reduced to 90% and 50% of the initial value under 530 and 940 nm, respectively (Supplementary Fig. 11). The relatively significant reduction under the NIR light can be attributed to the instability of Sn cations in the air⁴².

42. Ricciarelli, D. et al. Instability of tin iodide perovskites: bulk p-doping versus tin oxidation. *ACS Energy Lett.* **5**, 2787-2795 (2020).

Q5) It is not clear why the optoelectronic logic gate platform can suggest an integrated chip for optical computing, optical communication, logic memory units, and photonic quantum computation (in the conclusion section). More detailed explanation is necessary even though the authors simply mentioned the future plan.

[Answer]

We agree with the reviewer's comment that a more detailed explanation for the possible application is necessary.

This study would contribute to designing a highly integrated multifunctional perovskite optical devices in short-term. Moreover, concerning the capability of delicately modulating optoelectronic states by controlling the photocurrent polarity, the proposed array platform will pave the way for developing and advancing other applications such as light-fidelity (Li-Fi) transmission, security circuits, data processors, and healthcare sensors in the future.

1. The Li-Fi is emerging as a future wireless system beyond the conventional wireless fidelity (Wi-Fi) technology owing to high energy efficiency, compatibility within most households, fast data transmission, and enhanced security^{R20}. As a pivotal component for the system, photodetectors (PDs) merely serve as a recorder that converts photonic signals to electrical information and transfers it to exterior processing units. In the Li-Fi system, however, the SPPD-OELG platform can take over functions of the PD and execute the role of processor simultaneously, thus enabling efficient energy consumption and high-speed computing.
2. The capability of running multiple logics can realize a novel encryption technology. For instance, Wu et al. contrived the security circuit using a polymorphic logic device executing NAND and NOR^{R30}. In the same manner, assuming that the NIR and visible lights are perceived as data input (ciphertext) and gate modulator (encryption key), respectively, the SPPD-OELG can serve as a code reader to output a plaintext.
3. The data processor carrying out actual data processing under specific instructions (e.g., add, multiply, or count) can be implementable with a combinational circuit of multiple SPPD-OELG devices in a single chip, which is more spatially efficient compared to the conventional logic circuit based on electronic transistors.
4. The role of SPPD can be expanded to healthcare systems, monitoring bio-information using light sources. For example, blood oxygen saturation (SO₂), defined as the percent of

oxyhemoglobin in the total amount of hemoglobin in the blood, has been regarded as a significant biomarker for real-time monitoring of respiratory diseases. One recent tactic to measure the SO₂ is to track the reflectance change in the blood under the illumination of two different light sources (e.g., red and NIR range)^{R31}. Compared to existing systems requiring two separated photodetectors, a SPPD-based system can be more spatially efficient.

[Action]

We added new sentences in the discussion part on **page 15** of the revised manuscript as follows:

Manuscript (page 15, paragraph 2): The data processing under specific instructions (e.g., add, multiply, or count) could be implementable with a combinational circuit of multiple SPPD-OELGs in a single chip, which is much spatially and costly efficient compared to the conventional logic circuit based on electronic transistors, potentially advancing to future applications for optical computing, optical communication, and logic memory. In short-term view, this development can be applicable to light-fidelity (Li-Fi) transmission²⁰, security circuits⁴⁴, and healthcare sensors⁴⁵, utilizing the distinguished optoelectronic output states based on the photocurrent polarity.

44. Wu, P. et al. Two-dimensional transistors with reconfigurable polarities for secure circuits. *Nat. Electron.* **4**, 45-53 (2021).
45. Khan, Y. et al. A flexible organic reflectance oximeter array. *PNAS* **115**, E11015-E11024 (2018).

[References]

- R17. Dou, L. et al. Solution-processed hybrid perovskite photodetectors with high detectivity. *Nat. Commun.* **5**, 5404 (2014).
- R18. Chen, Z. et al. Solution-processed visible-blind ultraviolet photodetectors with nanosecond response time and high detectivity. *Adv. Opt. Mater.* **7**, 1900506 (2019).
- R19. Shen, L. et al. A self-powered, sub-nanosecond-response solution processed hybrid perovskite photodetector for time resolved photoluminescence-lifetime detection. *Adv. Mater.* **28**, 10794-10800 (2016).
- R20. Xiao, C. et al. Mechanism of electron-beam-induced damage in perovskite thin films revealed by cathodoluminescence spectroscopy. *J. Phys. Chem.* **119**, 26904-26911 (2015).
- R21. Reuter, P. et al. Electron beam-induced current (EBIC) in solution-processed solar cells. *Scanning* **33**, 1-6 (2011).
- R22. Ricciarelli, D. et al. Instability of tin iodide perovskites: bulk p-doping versus tin oxidation. *ACS Energy Lett.* **5**, 2787-2795 (2020).
- R23. Shi, L. et al. Accelerated lifetime testing of organic-inorganic perovskite solar cells encapsulated by polyisobutylene. *ACS Appl. Mater. Interfaces* **9**, 25073-25081 (2017).
- R24. Bella, F. et al. Improving efficiency and stability of perovskite solar cells with photocurable fluoropolymers. *Science* **354**, 203-206 (2016).
- R25. Hwang, I. et al. Enhancing stability of perovskite solar cells to moisture by the facile hydrophobic passivation. *ACS Appl. Mater. Interfaces* **7**, 17330-17336 (2015).
- R26. Kim, H. et al. Enhanced stability of MAPbI₃ perovskite solar cells using poly(p-chloroxylylene) encapsulation. *Sci. Rep.* **9**, 15461 (2019).
- R27. Lee, Y. I. et al. A low-temperature thin-film encapsulation for enhanced stability of a highly efficient perovskite solar cell. *Adv. Energy Mater.* **8**, 1701928 (2018).
- R28. Choi, E. Y. et al. Enhancing stability for organic-inorganic perovskite solar cells by atomic layer deposited Al₂O₃ encapsulation. *Sol. Energy Mater. Sol. Cells* **188**, 37-45 (2018).
- R29. Chang, C.-Y. et al. High-performance, air-stable, low-temperature processed semitransparent perovskite solar cells enabled by atomic layer deposition. *Chem. Mater.* **27**, 5122-5130 (2015).
- R30. Wu, P. et al. Two-dimensional transistors with reconfigurable polarities for secure circuits. *Nat. Electron.* **4**, 45-53 (2021).
- R31. Khan, Y. et al. A flexible organic reflectance oximeter array. *PNAS* **115**, E11015-E11024 (2018).

REVIEWERS' COMMENTS

Reviewer #1 (Remarks to the Author):

The authors have achieved a great revision. I recommend to accept it after citing the following related paper [Nature Electronics, 2021, 4, 631-632].

Reviewer #2 (Remarks to the Author):

I think the revised manuscript can be published as it is

Reviewer #3 (Remarks to the Author):

The revised manuscript by Kim and co-workers has adequately addressed the comments and issues raised in the initial review report. The newly added data support the conclusions. Thus, I think the revised manuscript will be suitable for publication in Nature Communications.

In details, I have still one concern (regarding my original question #2), which is about the rise and decay times of vertical perovskite-based photodetectors. Although it is impressive to reduce the rise and decay times by minimizing the device size, the rise and decay times in this work seem to be comparable to those in previous studies (as shown in the Supplementary Table 2). Therefore, I would like to suggest that the authors should tone down the overexpression such as "ultrafast" (page 4, line 84) and "very fast" (page 11, line 249).

Reviewer #1

The authors have achieved a great revision. I recommend to accept it after citing the following related paper [Nature Electronics, 2021, 4, 631-632].

[Answer]

We sincerely appreciate this comment. We delight in citing a great reference that provides the latest information on the bipolar photoresponse.

[Action]

We cited the suggested references including the reference 22 on **page 4** of the revised manuscript as follows:

Manuscript (page 4, paragraph 1): For instance, Yang et al. reported the wavelength-induced dual-polarity phenomenon from a p-n heterojunction diode composed of n-type ZnO nanowires and p-type thermoelectric thin film (SnS or Sb₂Se₃)¹⁹⁻²².

22. Yang, Y. Controlling photocurrent direction with light. *Nat. Electron.* **4**, 631-632 (2021).

Reviewer #2

I think the revised manuscript can be published as it is

[Answer]

We are grateful for the thoughtful assessment of our revised manuscript. The reviewer's previous comments substantially helped improve the quality of our manuscript.

Reviewer #3

The revised manuscript by Kim and co-workers has adequately addressed the comments and issues raised in the initial review report. The newly added data support the conclusions. Thus, I think the revised manuscript will be suitable for publication in Nature Communications.

In details, I have still one concern (regarding my original question #2), which is about the rise and decay times of vertical perovskite-based photodetectors. Although it is impressive to reduce the rise and decay times by minimizing the device size, the rise and decay times in this work seem to be comparable to those in previous studies (as shown in the Supplementary Table 2). Therefore, I would like to suggest that the authors should tone down the overexpression such as "ultrafast" (page 4, line 84) and "very fast" (page 11, line 249).

[Answer]

We agree with the reviewer's opinion. We deleted the overexpressed adjectives "ultrafast" and "very fast". Thank you again for this constructive comment.

[Action]

We deleted the overstated expression on **page 5** and **13** of the revised manuscript as follows:

Manuscript (page 5, paragraph 2): The back-to-back SPPD demonstrated a sensitive photoresponse and a substantial on-off ratio.

Manuscript (page 13, paragraph 1): This tuning technique (OR \leftrightarrow AND) using the NIR modulation was reversible and repeatable.